# From Mechanistic Interpretability to Mechanistic Biology: Training, Evaluating, and Interpreting Sparse Autoencoders on Protein Language Models

Etowah Adams [* 1]   Liam Bai [* 2]   Minji Lee [1]   Yiyang Yu [1]   Mohammed AlQuraishi [1]

## Abstract

Protein language models (pLMs) are powerful predictors of protein structure and function, learning through unsupervised training on millions of protein sequences. pLMs are thought to capture common motifs in protein sequences, but the specifics of pLM features are not well understood. Identifying these features would not only shed light on how pLMs work, but potentially uncover novel protein biology—studying the model to study the biology. Motivated by this, we train sparse autoencoders (SAEs) on the residual stream of a pLM, ESM-2. By characterizing SAE features, we determine that pLMs use a combination of generic features and family-specific features to represent a protein. In addition, we demonstrate how known sequence determinants of properties such as thermostability and subcellular localization can be identified by linear probing of SAE features. For predictive features without known functional associations, we hypothesize their role in unknown mechanisms and provide visualization tools to aid their interpretation. Our study gives a better understanding of the limitations of pLMs, and demonstrates how SAE features can be used to help generate hypotheses for biological mechanisms. We release our code, model weights and feature visualizer.

## 1. Introduction

Protein language models (pLMs) are language models trained on large datasets of protein sequences. In the process of minimizing loss on their pre-training task–usually masked token prediction–they learn representations of proteins useful for downstream tasks including protein structure (Lin et al., 2023) and function (Rao et al., 2019; Notin et al., 2022) prediction. For this reason, pLMs have become a growing component of the protein biologist's toolkit.

As these models are widely adopted, a number of studies have investigated their inner workings to understand their limitations. Previous work suggests pLMs do not learn the biophysics of proteins, but rather store common sequence motifs and contacts (Zhang et al., 2024). For many downstream tasks, pLM performance is driven by features learned in early layers and does not scale well with pre-training (Li et al., 2024). Yet exactly what pLMs have learned about proteins remains unknown.

Recent work in mechanistic interpretability points to a path forward. Sparse autoencoders (SAEs) have successfully extracted interpretable features from large language models (LLMs) like GPT-4 and Claude (Gao et al., 2024; Templeton et al., 2024). SAEs decompose model activations into a sparse, high-dimensional representation where individual latent dimensions often have interpretable activation patterns. These features provide a window into the inner workings of these complex models. Training SAEs on pLMs can serve an additional purpose. Unlike natural languages, which we intuitively understand, the language of proteins is more cryptic. SAE features from pLMs could potentially unveil patterns in protein sequences that we have not yet discovered.

In this work, we train SAEs on the residual stream of ESM-2, a widely used pLM (Lin et al., 2022) (Figure 1a). To interpret the SAE latents, we develop InterProt, a tool that visualizes latent activations on protein sequences and structures. We find that some features are highly interpretable, corresponding to a range of concepts from secondary structure elements to entire domains, as reported contemporaneously by Simon & Zou (2024). We conduct a human evaluation study and find that SAE latents are consistently rated as interpretable, in contrast to the ESM baseline. Furthermore, we find that many features activate highly only on specific protein families, suggesting that pLMs rely on internal representations of protein families to perform their pre-training task. We introduce a method to categorize SAE latents based on their family specificity and activation pat-

---
[*]Equal contribution   [1]Department of Systems Biology, Columbia University [2]Ginkgo Bioworks, Boston. Correspondence to: Etowah Adams <ea2901@columbia.edu>.

*Proceedings of the 42nd International Conference on Machine Learning*, Vancouver, Canada, PMLR 267, 2025. Copyright 2025 by the author(s).

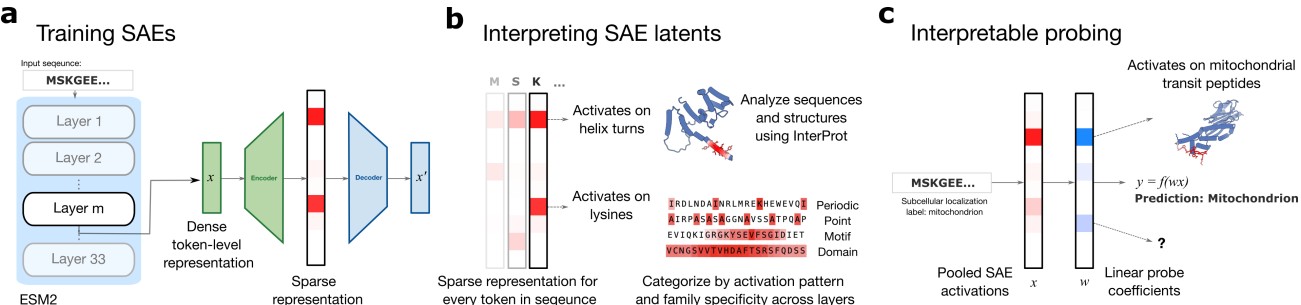

*Figure 1.* Overview of our paper. **a**. We train SAEs on the output of ESM intermediate layers. **b**. We interpret SAE latents using our latent visualizer InterProt and categorize features based on their activation pattern and family specificity. **c**. By interpreting the weights of linear models trained on SAE latents, we show how features which correspond to known sequence determinants can be identified.

tern (Figure 1b).

Having developed a framework for understanding features, we explore the impact of SAE hyperparameters and ESM layers. We find that increasing the sparsity imposed on SAE latents increases the number of features that are family-specific as does increasing the expansion factor. The number of family-specific features also varies substantially by ESM layer, with middle layers containing the most family-specific features.

With the goal of mapping SAE features to important protein properties, we train linear probes on SAE features and analyze the most predictive features. Our method identifies features corresponding to known sequence determinants for properties such as thermostability and subcellular localization. These results demonstrate that SAEs can help us not only understand pLMs, but also the data on which they are trained. By making pLM representations interpretable, we envision that SAE features may help generate hypotheses for unknown biological mechanisms (Figure 1c).

## 2. Related Works

### 2.1. Interpreting protein language models

Previous evaluations of pLM representations on downstream tasks provide evidence of features that pLMs may be learning (Rao et al., 2019; Dallago et al., 2021; Detlefsen et al., 2022; Li et al., 2024). In particular, Li et al. 2024 performs a comprehensive analysis of ESM performance on a range of downstream tasks across different model sizes and layers. Of the tasks they tested, they find that only structure-related prediction tasks scales with model size, and that most tasks use "low-level features" learned early in pre-training. A key limitation of using downstream tasks to understand the features learned by pLMs is that features cannot be discovered without supervision. In contrast, SAEs can uncover features in a more unsupervised manner.

pLMs seem to use a notion of sequence homology and recurring protein motifs. Using influence functions, Gordon et al. (2024) finds that pLM-derived sequence likelihoods are driven by homologous protein training data. Zhang et al. (2024) propose that pLMs store common motifs in proteins and the pairwise contacts between them. They arrive at this proposition by analyzing what residue segments must be unmasked to recover a sequence contact (via the "categorical Jacobian").

Many studies have explored how pLMs work mechanistically as well. Analyses of attention matrices in pLMs have been shown to resemble pairwise contact maps (Vig et al., 2021; Rao et al., 2021; Lin et al., 2023), and contain information such as binding sites (Vig et al., 2021) and allosteric sites (Kannan et al., 2024; Dong et al., 2024). They have been comprehensively mapped to Gene Ontology terms (Chen et al., 2025).

Concurrent to our work, Simon & Zou (2024) trained SAEs on ESM-2 and presented methods to analyze their latents: visualizing them, evaluating them against Swiss-Prot annotations, and interpreting them with a language model. Our work builds on this foundation with a different SAE architecture and evaluation methods that focus on coevolution, human preference, and downstream tasks (subsection A.3).

### 2.2. Dictionary learning for feature extraction from models trained on biological data

Following the success of extracting interpretable features from LLMs, a number of works have applied sparse dictionary learning techniques to models trained on scientific data. Donhauser et al. (2024) evaluates the possibility of extracting features from models trained on microscopy data. Interpretable features have also been uncovered from single cell foundation models (Schuster, 2024; Adam Green, 2024). Associating the extracted features with functional labels via probing, however, has not yet been explored in

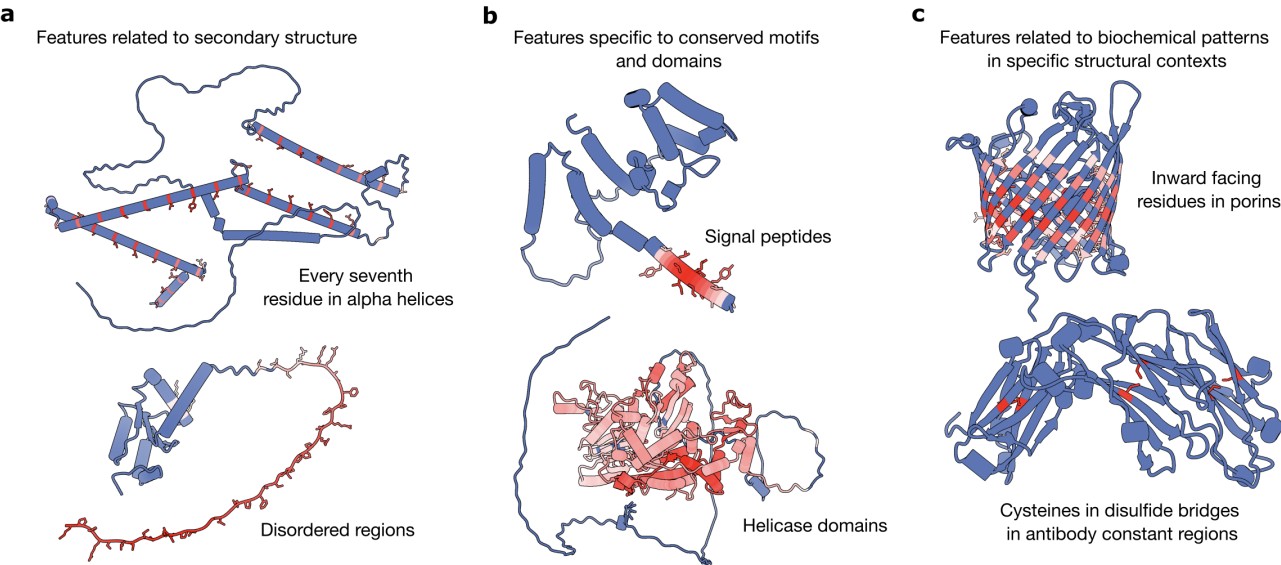

*Figure 2.* Examples of SAE features. We find features related to secondary structure (a), conserved motifs and domains (b), and biochemical patterns in specific structural contexts (c). The structures are colored according to activation (red: activation, blue: no activation).

this context.

## 3. Background and Methods

### 3.1. Sparse autoencoders (SAEs)

An SAE is an autoencoder designed to learn efficient and meaningful representations by enforcing sparsity constraints on encoder output. We use SAEs with TopK activation to enforce sparsity, as introduced in Gao et al. (2024). This activation function only allows the $k$ largest latents to be non-zero. We choose TopK SAEs due to their improved reconstruction at a given level of sparsity compared to other techniques, and because it allows us to directly set the L0-norm of the encoding using $k$. We note that improved reconstruction may come at the cost of increased feature absorption (Karvonen et al., 2024).

The encoder and decoder are defined as:

$$z = \text{TopK}\left(W_{\text{enc}}\left(x - b_{\text{pre}}\right)\right)$$

$$\hat{x} = W_{\text{dec}}z + b_{\text{pre}}$$

where $W_{\text{enc}}$ projects the residual stream into the SAE latent space and $W_{\text{dec}}$ the reverse. TopK zeros all latents that are not in the top $k$. The loss is simply the reconstruction mean squared error $\mathcal{L} = \|x - \hat{x}\|_2^2$. We train our TopK SAEs on 1 million random sequences under 1022 residues from UniRef50 (Suzek et al., 2007).

Following Lieberum et al. (2024), we refer to the hidden dimensions of the SAE as "latents" to clearly distinguish them from the underlying conceptual features of the model. This is in contrast to previous works which use the word feature to refer to both (Bricken et al., 2024).

### 3.2. Downstream tasks

**Secondary structure.** The secondary structure dataset from TAPE (Rao et al., 2019) contains residue-level labels in 3 classes: alpha helix, beta strand, or other. To prevent data leakage, sequences are divided into training and test sets based on a threshold of 25% sequence identity.

**Subcellular localization.** Subcellular localization is a protein-level label for where a protein is localized within a cell. The dataset from Almagro Armenteros et al. (2017) consists of UniProt-sourced eukaryotic proteins, with stringent homology-based train-test splits.

**Thermostability.** The thermostability dataset from the Melteome Atlas (Jarzab et al., 2020) measures the melting temperature of 48,000 proteins across 13 species. We use the train-test split from FLIP (Dallago et al., 2021) containing all sequences clustered at 25% sequence identity.

**Mammalian cell expression.** The dataset curated by Masson et al. (2022) contains expression data of 2,165 proteins from the human secretome in Chinese hamster ovary (CHO) cells, commonly used mammalian hosts for therapeutic protein production. We use this dataset as a binary classification task to determine whether a protein can be successfully expressed in CHO cells. We split proteins into training and

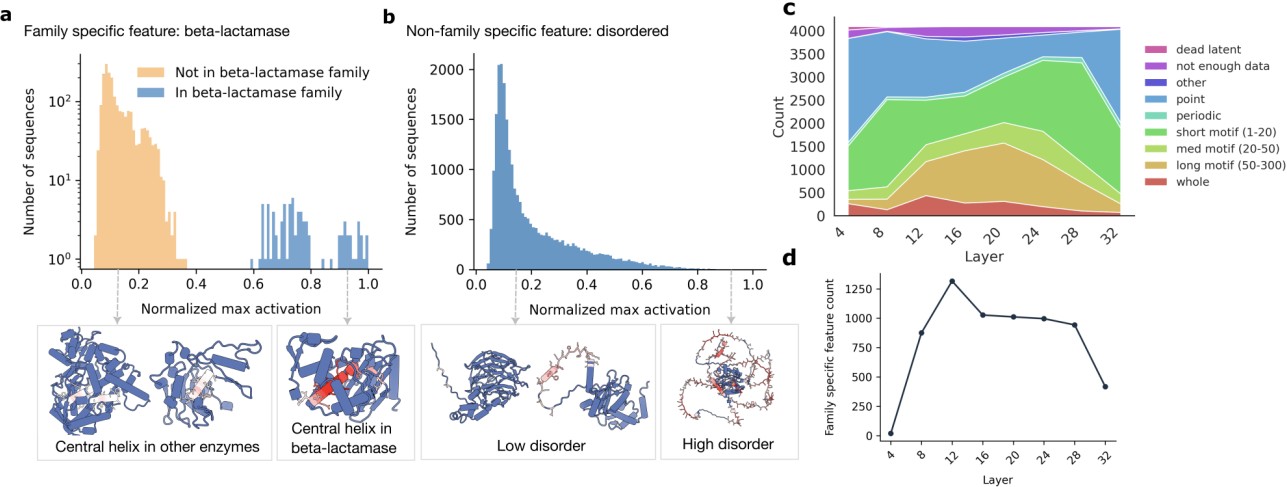

*Figure 3.* **a**. Example of a family specific feature, the central helix of a beta lactamase. The histogram shows the distribution of maximum activations in the sequences which activate the latent. **b**. Example of a non-family specific feature, corresponding to disordered sequences. **c**. Latents classified by activation pattern across layers. **d**. Number of family-specific latents across layers. The SAEs used in c and d have a hidden dimension of 4096 and k=64.

test sets based on a threshold of $40\%$ sequence identity.

### 3.3. Linear probes

For each task and for varying ESM layers, we train linear probes on both ESM and SAE embeddings. We mean-pool embeddings over the length of the sequence for tasks with protein-level labels: subcellular localization, thermostability, and mammalian cell expression. For all linear regression and logistic regression tasks, we perform grid search over a range of regularization strengths and pick the best model using a validation set. See Table 1 for a summary of the probe implementation for each task.

### 3.4. Manual feature interpretability study

Following Rajamanoharan et al. (2024), we conducted a blinded study in which seven participants (graduate students and undergraduates familiar with protein biology) rated the interpretability of SAE and ESM latents from layer 24. Embeddings were computed on sequences from Swiss-Prot clustered at 30% sequence identity (see subsection A.6). Using our InterProt visualizer, each participant viewed 100 randomly selected features (drawn uniformly from both models) and rated their interpretability as *yes*, *no*, or *maybe*.

## 4. Feature Analysis

### 4.1. SAEs uncover interpretable features

We train TopK SAEs on the residual stream of different layers of ESM-2 (650M) to explore its learned representations.

To interpret a given SAE latent dimension (*i.e.*, a latent), we examine the top activating protein sequences, *i.e.*, those that produce the largest activation for that dimension. We developed a latent visualizer, InterProt, to streamline the process of identifying features. InterProt allows users to align the most-activating sequences and highlight their corresponding protein structure by activation strength. It is open source and can be accessed at `https://interprot.com`.

Through these visualizations, we gain insight into how ESM internally represents protein sequences (Figure 2). Many discovered features correspond to recognizable biological concepts, such as secondary structure elements (*e.g.*, alpha helices and beta strands), short conserved motifs, or entire functional domains. Some features capture biochemical concepts like cysteine-cysteine bonds or the orientation of residue side chains. In many ways, latents capture characteristics of a protein that a biologist might notice.

Notably, while many latents are intuitive (*e.g.*, focused on conserved motifs), others are unexpectedly specific. For instance, rather than a single, generic "alpha helix" latent, we observe multiple helix-related latents that activate under more context-dependent conditions (*e.g.*, at the beginning of a helix, only when the helix is buried, or when the helix meets a certain length criterion). Other examples include highly context specific amino acids, such as asparagine residues in disordered regions.

To systematically evaluate interpretability, we conducted a blinded human study to rate SAE and ESM latents (subsection 3.4). SAE latents were rated as interpretable far more

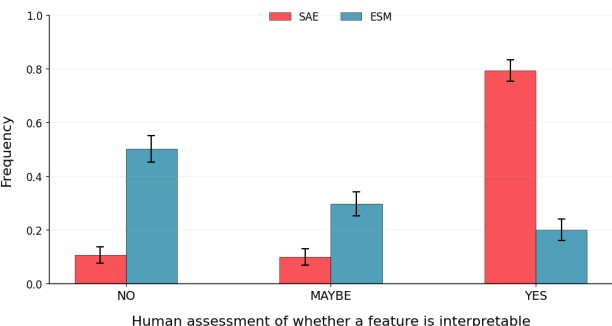

*Figure 4.* Human rater scores of SAE or ESM latents from layer 24. The error bars indicate the 95% confidence interval, assuming a binomial 1-vs-all distribution.

often than those from ESM (Figure 4). Approximately 80% of SAE features received a "yes" rating for interpretability, broadly consistent with previous findings on SAEs trained on language model activations (Rajamanoharan et al., 2024).

### 4.2. SAE latents reveal a learned notion of protein families

Intriguingly, a large subset of latents appear to be protein family-specific. These latents activate weakly on most sequences but strongly on proteins belonging to a particular family. For example, the latent shown in Figure 3a appears to highly activates only for the central helix in beta-lactamases, but only exhibits marginal activations when a similar structural motif appears in proteins outside the beta-lactamase family.

By contrast, a non-family-specific latent such as the "disordered" latent in Figure 3b, responds broadly to intrinsically disordered regions in proteins across many families. These observations collectively suggest that pLMs learn internal representations that, in part, mirror known sequence-homology based groupings of proteins, akin to a model-internal notion of protein families.

To better understand the relationship between a family-specific latent and its associated protein family, we investigate what happens when we "steer" ESM using the latent. Specifically, we set the activation of a family-specific latent to a multiple of its maximum activation, and continue the model's forward pass using the reconstructed activation. We find that, compared to steering random latents which activate on a sequence, steering family specific latents causes fewer residues to change (Figure 7). Investigating how to steer the model in useful ways remains an open direction for future work.

### 4.3. Classifying latents

Motivated by these observations, we propose two classification schemes for the SAE latents.

**Activation Pattern.** Latents exhibit distinct activation patterns across their top sequences (Figure 1b). To capture these differences, we use a rule-based scheme that classifies features into categories such as: point (activates on single residues at a time), periodic (activates in a regular interval, repeating every n residues), motif (activates in short, medium, or long contiguous intervals), and domain (activates over nearly the entire sequence). Refer to Table 2 for precise definitions.

**Family Specificity.** To quantify family specificity, we use each latent for binary classification: given a protein, can this latent's activations be used to predict membership in a particular family? We label a latent "family-specific" if it achieves an F1 score $> 0.7$ at a certain activation threshold. We use all protein sequences from Swiss-Prot (Boeckmann et al., 2003) (clustered as 30% sequence identity) and categorized into InterPro protein families (Paysan-Lafosse et al., 2023) for our evaluation. Refer to subsection A.6 for details.

These classifications allow us to systematically group SAE features and draw broader conclusions about their roles in representing protein structure and function.

### 4.4. Influence of SAE training choices on latent classification

Prior literature on LLMs suggests that SAE-learned features depend on multiple factors, including model hyperparameters (*e.g.*, the sparsity level and latent dimension size) and training data (*e.g.*, which layer's activations are used). We therefore examine how these factors impact the types of features that emerge in pLMs.

**Effect of k.** In TopK SAEs, the $k$ hyperparameter controls how many latents can be active for a given input, effectively setting the sparsity level. We find that, for a fixed hidden dimension, lower $k$ (*i.e.*, higher sparsity) yields more family-specific features (Figure 8a). One plausible explanation is that family-specific features constitute the most salient signals used in reconstruction.

**Effect of expansion factor.** We also experiment with increasing the latent dimension, called the expansion factor. While we initially hypothesized that a larger latent dimension might shift the distribution of feature activation patterns, we observed that activation pattern classifications (point, motif, domain, etc.) remain largely consistent, while increasing the number of family specific features (Figure 8b).

**Comparison across layers.** Finally, we investigate layer-wise differences in the learned features. Previous studies have shown that downstream performance can vary substantially depending on which layer representations are used, suggesting that different layers capture different degrees of abstraction. In our experiments, family-specific features tend to peak in certain early-to-mid layers of ESM, then

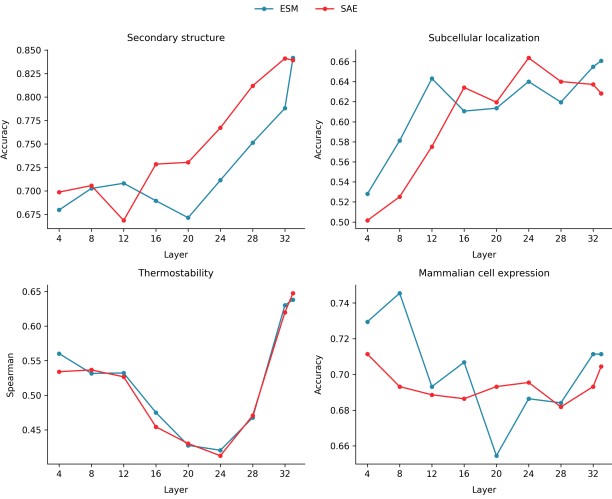

*Figure 5.* Linear probe performance on ESM vs. SAE embeddings across ESM layers. In all 4 downstream tasks, SAE performs competitively with ESM. For secondary structure prediction, SAE consistently outperforms ESM in most layers. All classifications tasks are measured using accuracy; thermostability is measured using Spearman's rank correlation.

decline in later layers (Figure 3d). Similarly, features which activate contiguously over long sections (motif/domain) are more common in earlier layers, whereas later layers exhibit features with shorter contiguous activations (Figure 3c). Given that later layers are thought to specialize for final logit computation (Gao et al., 2024), one possibility is that shorter, more specific activations better serve this goal.

## 5. Interpretable Probing

Representations learned by pLMs can be used in downstream tasks. Often, this is as simple as training a linear model on the pLM embeddings. Although this leads to a predictive model, it does so at the expense of interpretability, as the coefficients of the linear model do not correspond to interpretable features. In this section, we explore the ability of SAE representations to overcome this limitation. Since SAE latents are often interpretable, those that contribute significantly to prediction could reveal biological reasons that explain the prediction.

### 5.1. SAE probes perform competitively with ESM probes

In the context of LLMs, linear probes on SAE embeddings can achieve performance similar to, or in some cases better than, linear probes on their base model embeddings (Bricken et al., 2024; Kantamneni et al., 2024). To explore this dynamic in pLMs, we train linear probes on ESM and SAE embeddings across different layers. Following Li et al. (2024), we evaluate the probes on benchmarked downstream tasks:

secondary structure, subcellular localization, thermostability, as well as a novel task, mammalian cell line expression prediction.

We find that linear probes on SAEs achieve performance similar to their ESM baselines across all layers (Figure 5). For secondary structure prediction, the SAE probe consistently outperforms the ESM probe, a result we discuss in the next section and attribute to the abundance of SAE latents that correspond to secondary structure.

### 5.2. SAE probes uncover interpretable latents corresponding to known mechanisms

To assess the interpretability of linear probes trained on SAE latents, we analyze the distribution of their coefficients (Figure 10) and manually inspect the latents with the largest coefficients in the InterProt visualizer [1]. We find that those latents often correspond to biological concepts with known relevance to each task (Figure 6). [2]

#### 5.2.1. SECONDARY STRUCTURE

Secondary structure prediction is a task that is known to utilize the features learned during pLM pre-training (Li et al., 2024). Our analysis of the highest-weighted SAE latents in our secondary structure probe confirms this and provides an explanation for why early layers perform worse.

Our SAE probes reveal that most layers contain features corresponding to secondary structure elements. The most positive coefficients for the alpha helix class are typically latents that activate exclusively on helices (Figure 6a). They range from specific helix residues, to entire helices, to helix-helix interactions. We observe similar results for the two other classes: beta strand and *other*, where *other* tends to correlate with features that activate on disordered regions.

Why is classification performance worse in early layers? We find that early layers do not contain generic features for secondary structure. For example, the top latent for helix classification in layer 4 (L4/2624) detects proline (P) residues, which often appear at helix boundaries due to their disruptive effect on hydrogen bonding. This reliance on a weak amino acid-level correlation highlights the absence of a more generalized helix feature. Similarly, the top latent for the disordered classifier activates on N-terminal residues, reflecting the spurious correlation that many proteins have disordered regions at their N-terminus, rather than capturing

---

[1] Linear probe coefficients are not comparable across tasks because different tasks may apply different regularization strengths, optimized via grid search.

[2] In this section and in Figure 6, we discuss latents from a range of layers. The choice of layers is qualitative—primarily based on latents we find particularly interpretable or notable—and does not imply the lack of interesting latents in undiscussed layers.

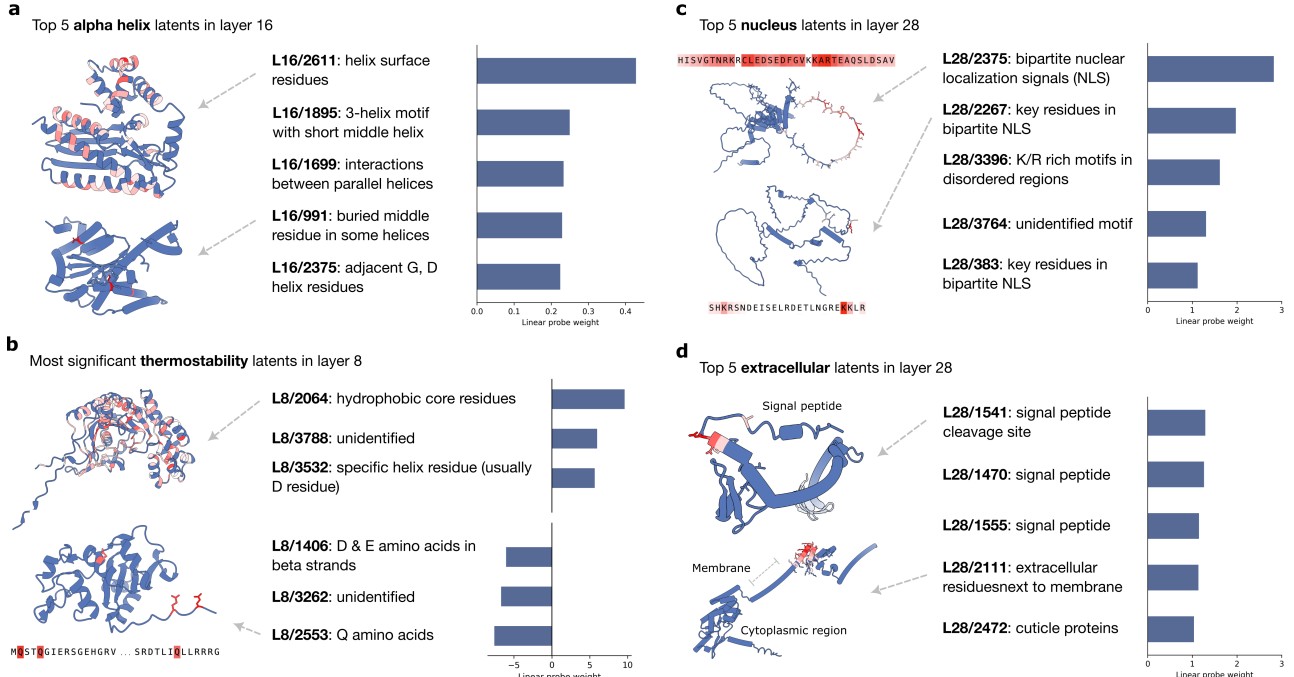

*Figure 6.* Linear probes identify predictive SAE latents for downstream tasks. **a**. The top 5 alpha helix latents from the secondary structure prediction task. These latents activate mostly exclusively on alpha helices (SAE trained on ESM layer 16). **b**. The latents corresponding to the 3 most positive and 3 most negative coefficients from the linear probe on thermostability. The most positive latent activates on residues in the hydrophobic core, which is structurally important for thermostability; the most negative latent activates on glutamine (Q) amino acids, whose absence is correlated with increased thermostability (SAE trained on ESM layer 8). **c-d**. Top 5 nucleus and extracellular latents from the subcellular localization prediction task. The top nucleus latents activate on bipartite nuclear localization signals (NLS), which contain a variable-length linker and generally follow the pattern $R/K(X)_{10-12}KRXK$. The top extracellular latent activates on signal peptide cleavage sites, while another top latent activates on the extracellular regions adjacent to the membrane (SAE trained on ESM layer 28).

fundamental structural properties.

### 5.2.2. SUBCELLULAR LOCALIZATION

Transport mechanisms involving sequence motifs such as transit peptides and localization signals determine the location of proteins within the cell. We find that linear probes trained on SAE latents can uncover some of these known mechanisms.

**Nuclear localization.** The most highly weighted latents for nuclear localization often correspond to nuclear localization signals (NLS). For example, the top latents in the probe for layer 28 probe activates on K/R rich sequence motifs (Figure 6c) and on bipartite nuclear localization signals (L28/2375), which generally follow the pattern $R/K(X)_{10-12}KRXK$ (Lu et al., 2021). The latent activates strongly on the R/K motifs at each end and can flexibly recognize the variable-length linker sequence that connects them. These sequence motifs engage cellular mechanisms that enable the protein to enter the nucleus. As many NLS

variants still remain unknown, our SAE-based approach has the potential to become a novel method to aid their discovery and characterization.

**Extracellular localization.** The most positive coefficients for the extracellular class relate to signal peptides (Figure 6d). Signal peptides are short sequences at the N-terminus that trigger the secretory pathway to transport proteins outside the cell. L28/1541 activates on the signal peptide cleavage site while L28/1470 and L28/1555 activate across the peptide. The top latents also include a latent that activates on the extracellular portion of membrane-bound proteins (L28/2111), demonstrating precise recognition of transmembrane regions, as well as a latent (L28/2472) that only activates on cuticle proteins, extracellular proteins that support the exoskeleton of insects.

For the other localization classes, we also find a variety of interpretable latents, including mitochondrial transit peptides (L28/3277) and transmembrane regions (L28/2818, L28/1298) (Figure 9c). Overall, this task uncovered many

latents related to signal peptides, consistent with the observations from (Almagro Armenteros et al., 2017) that the attention maps of their transformer model trained on this task focus on sequence termini.

### 5.2.3. THERMOSTABILITY

Thermostability prediction is a task that does not scale with increased pre-training and larger models (Li et al., 2024). Our SAE probes reveal why—thermostability prediction relies most on simple amino acid statistics (Figure 6b).

Unlike the other tasks, the middle layers perform worse than early or late layers. We find that in middle layers the most significant latents tend to recognize specific amino acids. For example, layer 24's most positive latents correspond to Arginine (R), Tyrosine (Y), and Leucine (L); its most negative latents recognize Glutamine (Q), Aspartic acid (D), and Threonine (T). The absence of Glutamine is known to correlate with increased thermostability (Farias & Bonato, 2003).

In contrast, earlier layers seem to contain more relevant features. For instance, the most positive coefficient in layer 8 (L8/2064) corresponds to a latent that activates primarily on hydrophobic residues, which are important for stability (Figure 6b).

### 5.2.4. MAMMALIAN CELL EXPRESSION

We were interested to see if our SAE probing method could be used to help interpret a novel task with relevance to drug development: binary human protein expression in CHO cells. The two latents most predictive of expression activate on terminus-specific long motifs (Figure 9b). The latent most predictive of failed expression recognizes an ATP binding site, suggesting that its interactions with ATP may disrupt essential metabolic processes in the host cell.

While we found many latents hard to interpret, this task demonstrates the potential of our method in a practical setting. If SAEs can help discover biological mechanisms that affect expression in CHO cells, then can guide us in designing protein therapeutics with better developability profiles.

## 6. Discussion

In this work, we train sparse autoencoders (SAEs) to extract interpretable features from a protein language model (pLM). We categorize SAE latents based on their activation patterns and specialization across protein families, systematically analyzing how SAE sparsity, width, and layer choice affect feature extraction. Our work offers a new perspective on previous efforts to understand pLMs, confirming known results such as the central role of sequence homology (Lin et al., 2022; Gordon et al., 2024), the use of motifs (Zhang et al., 2024), and the unexpected effectiveness of early layers on some downstream tasks (Li et al., 2024). To facilitate further exploration, we introduce InterProt, an open-source visualizer for interpreting SAE latents.

Mechanistic interpretability methods like SAEs hold promise not only for understanding how complex models function but also for advancing scientific discovery (Donhauser et al., 2024). However, existing approaches often rely on matching features to concepts already known to the scientific community, potentially overlooking novel features. To address this gap, we developed a probing strategy that associates latents with candidate functional roles by training linear probes on SAE latents. Through this process, we uncover features that align with known sequence determinants, such as nuclear localization signals, which our probe identified as strongly predictive of nuclear localization. Had the role of these signals been unknown, our probing results would have provided a strong hypothesis for subsequent validation. More broadly, probing SAE latents offers a means to explain how pLMs achieve their performance on downstream tasks. Notably, we find that thermostability prediction relies heavily on latents which provide simple amino acid composition statistics.

In our probing experiments, we observe that not all predictive latents are easily interpretable. For instance, a predictor for membrane localization, latent L28/3154, predominantly activates on poly-alanine sequences, whose functional relevance remains unclear. Such hard-to-interpret but predictive features could result from biases in training data, limitations of our SAE, or ESM. However, another intriguing possibility is that these features correspond to biological mechanisms that have yet to be discovered.

**Limitations.** Our study has several limitations. First, we focus exclusively on the 650M-parameter variant of ESM-2, and extending our analysis to other model sizes, checkpoints, and architectures could yield richer insights into how features vary with pre-training scale and model architecture. Second, the features uncovered by our SAEs are sensitive to the training data distribution; although we use a diverse set of sequences from UniRef50, different datasets may produce a distinct feature set. Third, our probing approach relies on mean-pooling across the sequence for protein-level tasks–a strategy known to have shortcomings (NaderiAlizadeh & Singh, 2024)–and exploring alternative pooling methods may yield stronger associations between latents and functional properties. Finally, our capacity to interpret latents is inherently constrained by the current state of biological knowledge, leaving open the possibility that some of our interpretations, particularly for latents showing complex activation patterns, may be incomplete or misstated.

**Future directions.** The success of our SAE linear probes

in uncovering known biological mechanisms suggests their potential for discovering new ones. A key next step is to formulate and experimentally test hypotheses around unexplained predictive features. Since SAE-derived features depend on the dataset (Kissane et al., 2024), investigating how data diversity influences feature emergence could be valuable–training on task-relevant sequences may yield more function-specific latents. Another promising direction is model steering via SAEs: while Simon & Zou 2024 provides a proof of concept, further research is needed to determine whether SAEs can be used as a practical tool for protein engineering.

## Impact Statement

This paper presents work whose goal is to better understand protein language models. Our findings may aid in biological discovery, particularly in understanding sequence-function relationships. While our work has potential applications in protein engineering and drug discovery, we do not foresee any immediate ethical concerns beyond those generally associated with machine learning research.

## Acknowledgments and Funding

We thank Elana Simon, Neil Thomas, Andre Cornman, Yunha Hwang, Garyk Brixi, Peter Koo, Samuel Maffa, Philipe Chlenski, and Neel Nanda for helpful discussion around understanding and analyzing SAE latents. We thank Diego del Alamo, Daniel Saltzberg, James Michael Krieger, and many others on Twitter/X for contributing to SAE latent interpretation. We thank Pranay Satya for discussion on training SAEs. We thank the Evolved 2024 hackathon for providing compute resources for an earlier version of this project. We thank Zeming Lin, Uri Laserson, Noah MacCallum, Tess van Stekelenburg, and Matthew Nemeth for their feedback and support during the hackathon. M.L. and M.A. are supported by NIH grant R35GM150546.

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

## Author contributions

E.A. and L.B trained the SAE models, built the InterProt visualizer, and interpreted SAE latents. E.A. led the feature characterization experiments. L.B led the interpretable probing experiments. M.L. wrote the code for SAE steering and performed the family-specific latent steering experiments. Y.Y. contributed to the training code. E.A., L.B., M.A., provided feedback on all experiments. E.A. and L.B. wrote the manuscript with feedback from M.L. and M.A. All authors read and approved the manuscript.

## Competing interests

M.A. is a member of the scientific advisory boards of Cyrus Biotechnology, Deep Forest Sciences, Nabla Bio, and Oracle Therapeutics.

# A. Appendix

## A.1. Code availability and model weights

Our SAE training and evaluation code is publicly available at `https://github.com/etowahadams/interprot`. The feature visualization tool, *InterProt*, can be accessed at `https://interprot.com`, and pre-trained SAE weights are hosted at `https://huggingface.co/liambai/InterProt-ESM2-SAEs`.

## A.2. Background on protein language models

Protein language models (pLMs) apply techniques from natural language processing to model biological sequences, leveraging the ability of transformer architectures to learn patterns from large-scale data. Just as language models capture structure in text, pLMs can learn meaningful representations of proteins from large protein sequence databases. The ESM family of pLMs are BERT-style transformers trained with a masked language modeling objective to learn contextual embeddings of amino acids (Rives et al., 2021; Lin et al., 2023). In this work, we focus on the 650M-parameter variant of ESM-2 (Lin et al., 2023) which has 33 total layers.

## A.3. Key differences from InterPLM

Concurrent to our work, Simon & Zou (2024) also trained SAEs on ESM-2 models. Some key differences from work are as follows:

**SAE Architecture.** Simon & Zou (2024) used ReLU-based SAEs, we employ TopK SAEs, which allow explicit control over L0 sparsity.

**Evaluation.** Simon & Zou (2024) proposed an automated evaluation framework based on Swiss-Prot annotations, establishing that more SAE latents correspond to annotations compared to ESM neurons. We support this finding with a manual evaluation where participants blindly rated SAE latents and ESM neurons based on their interpretability in our visualization tool.

**Analysis.** Simon & Zou (2024) used an LLM to generate SAE feature descriptions based on activation examples and annotations. They also explored the application of SAE features in identifying missing annotations. Our analyses focused on classifying SAE latents based on activation patterns and family-specificity and using linear probes to uncover features most predictive in downstream tasks.

**Visualization.** Both works provide an interactive visualizer for SAE latents. interplm.ai from Simon & Zou (2024) display UMAP visualizations, results from Swiss-Prot concept mapping, and other statistics. Our visualizer, interprot.com, focus on the top activating sequences of each latent, displaying activation patterns overlaid on structure, sequence alignments, and information on shared protein family. We used this interface to conduct our human ratings experiment.

## A.4. Probe details

Table 1 summarizes the linear probes used in each downstream task.

*Table 1.* Summary of downstream tasks for interpretable probing

| DATASET | TASK TYPE | LABEL LEVEL | EVALUATION METRIC | PROBE IMPLEMENTATION |
|---|---|---|---|---|
| SECONDARY STRUCTURE | CLASSIFICATION | RESIDUE | ACCURACY | PYTORCH LINEAR CLASSIFIER |
| SUBCELLULAR LOCATION | CLASSIFICATION | PROTEIN | ACCURACY | SKLEARN LOGISTIC REGRESSION |
| THERMOSTABILITY | REGRESSION | PROTEIN | SPEARMAN'S $\rho$ | SKLEARN RIDGE REGRESSION |
| CHO CELL EXPRESSION | CLASSIFICATION | PROTEIN | ACCURACY | SKLEARN LOGISTIC REGRESSION |

## A.5. Steering family-specific latents

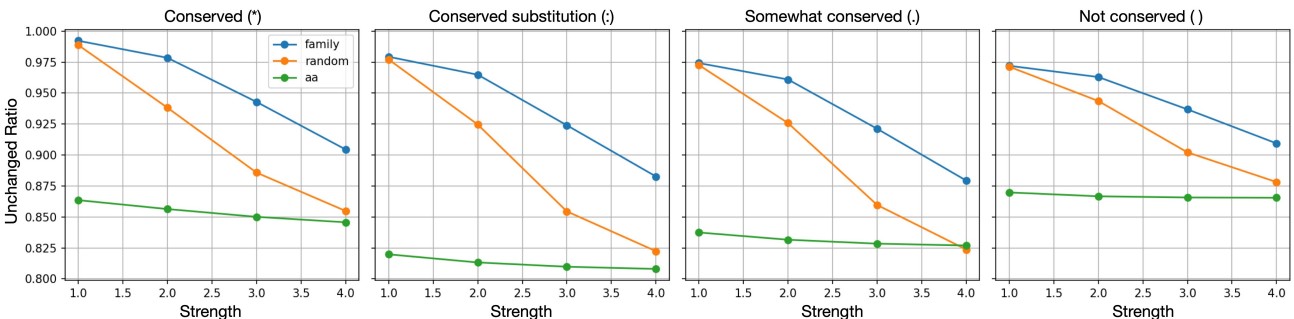

*Figure 7.* The fraction of residues that remains unchanged after steering, averaged for 2089 family specific features revealed using SAE.

We steer each family-specific latent and analyze the resulting sequence changes compared to steering random latents or latents corresponding to amino acid features. Steering refers to clamping of the activation of a specific latent to a fixed level–either increasing it to amplify the feature or decreasing it to suppress it. For each family-specific latent corresponding to an InterPro family, we identify the top-activating sequences, conduct protein-protein BLAST (Ye et al., 2006), and align sequences within the same InterPro family to obtain a multiple sequence alignment using Clustal Omega (Sievers et al., 2011). Finally, we determine the maximum activation of a family-specific latent and clamp the activation of the activating residues to 1×, 2×, 3×, and 4× the maximum value. As a baseline, we steer the residues which activated the family-specific latent with two random latent dimensions (1 and 11) and two amino acid specific latent dimensions (3267 for alanine and 3830 for glycine in the SAE with a hidden dimension of 4096 from layer 24). Because we choose two latents as representative random and amino acid latents, we take the average of the unchanged ratio. The unchanged ratio is the fraction of residues across the whole sequence that remains unchanged after steering.

Figure 7 compares the fraction of residues that remains unchanged after steering of family-specific (labeled "family"), random (labeled "random"), and amino acid (labeled "aa") latents, by the degree of conservation (decreasing order, left to right). For any degree of conservation, steering a family-specific latent has the least effect in the decoded sequence for the same steering strength, suggesting that family-specific latents capture underlying evolutionary constraints that resist disruption. The difference in unchanged ratio between family-specific and random latents is larger in conserved residues, implying steering family-specific latents tends to keep the residues with high consensus. Still, it is interesting that absolute unchanged ratio is roughly in a similar range for any degree of conservation.

## A.6. Classifying latents by family specificity

We evaluate the ability of a single SAE latent to classify a protein as belonging to a specific InterPro-defined protein family (Paysan-Lafosse et al., 2023). To do this, we determine the optimal normalized activation threshold that maximizes the F1 score for binary classification of protein families. Specifically, we sweep threshold values from 0.1 to 0.9 in 0.1 increments and select the threshold that results in the highest F1 score. Our final evaluation is conducted on a held-out test set.

For our evaluation, we use protein sequences from Swiss-Prot (Boeckmann et al., 2003). To reduce sequence redundancy, we cluster all Swiss-Prot sequences with a length of 1022 residues or fewer at 30% sequence identity using MMseqs2 (`mmseqs easy-cluster swissprot.fasta session tmp --min-seq-id 0.3 -c 0.9 -s 8 --max-seqs 1000000 --cluster-mode 1`) (Steinegger & Söding, 2017). Given the low sequence identity between clustered sequences, we perform a random train-test split for classification evaluation.

To determine which protein family to assess for each latent, we identify the InterPro family of the sequence that exhibits the highest activation for that latent. The F1 score is then computed for this family using the thresholding approach described above.

## A.7. Classifying latents by activation pattern

*Table 2.* Activation pattern classification criteria

| Category | Criteria |
| --- | --- |
| Dead Latent | If the latent is never activated by any test sequences, it is classified a dead latent. |
| Not Enough Data | If less than 5 sequences activate the latent then we say there is not enough data. |
| Periodic | Features that exhibit consistent activation patterns at regular intervals. These features must satisfy: (1) a high frequency of activation at specific positions (over 50% of distances between activations are the same two values), (2) a large number of activation regions (there are more than 10 activations per sequence), and (3) relatively short contiguous activation spans (median length of the top activating contig is less than 10). |
| Point | Features that activate in a highly localized manner, defined by a single, prominent activation site (the median length of the highest activating region is 1). |
| Motif (Short: 1-20) | Features that activate in short contiguous regions (median length of the highest activating region is $> 1$ and $< 20$) and have an overall mean activation coverage of less than 80%. |
| Motif (Medium: 20-50) | Features that activate in short contiguous regions (median length of the highest activating region is $\geq 20$ and $< 50$) and have an overall mean activation coverage of less than 80%. |
| Motif (Long: 50-300) | Features that activate in short contiguous regions (median length of the highest activating region is $\geq 50$ and $< 300$) and have an overall mean activation coverage of less than 80%. |
| Whole | Features that are active across nearly the entire sequence (overall mean activation coverage of greater than 80%.). |
| Other | Features that do not meet any of the above criteria are classified as "other." |

## A.8. Testing SAE hyperparameters

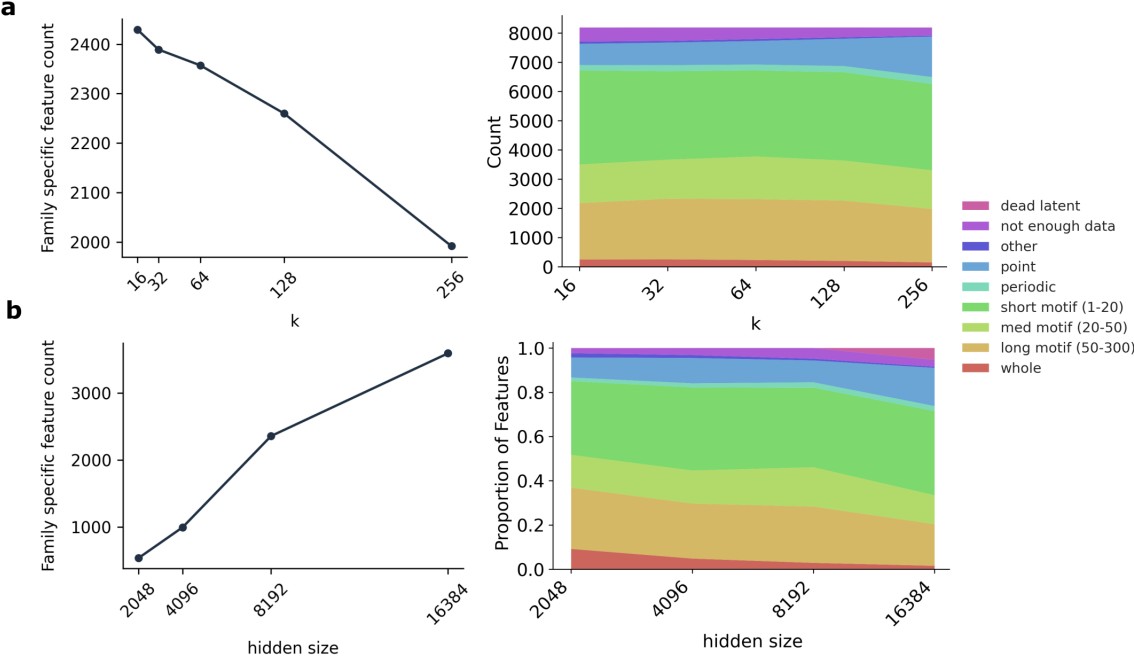

*Figure 8.* Testing SAE hyperparameters. **a.** $k$ parameter sweep (other hyperparameters held constant, layer=24, hidden size=8096) showing the number of family specific features (left) and the features categorized by activation pattern (right). **b.** SAE hidden size hyperparameter sweep (other hyperparameters held constant, layer=24, k=64) showing the number of family specific features (left) and the features categorized by activation pattern (right).

## A.9. Additional latent visualizations from linear probes

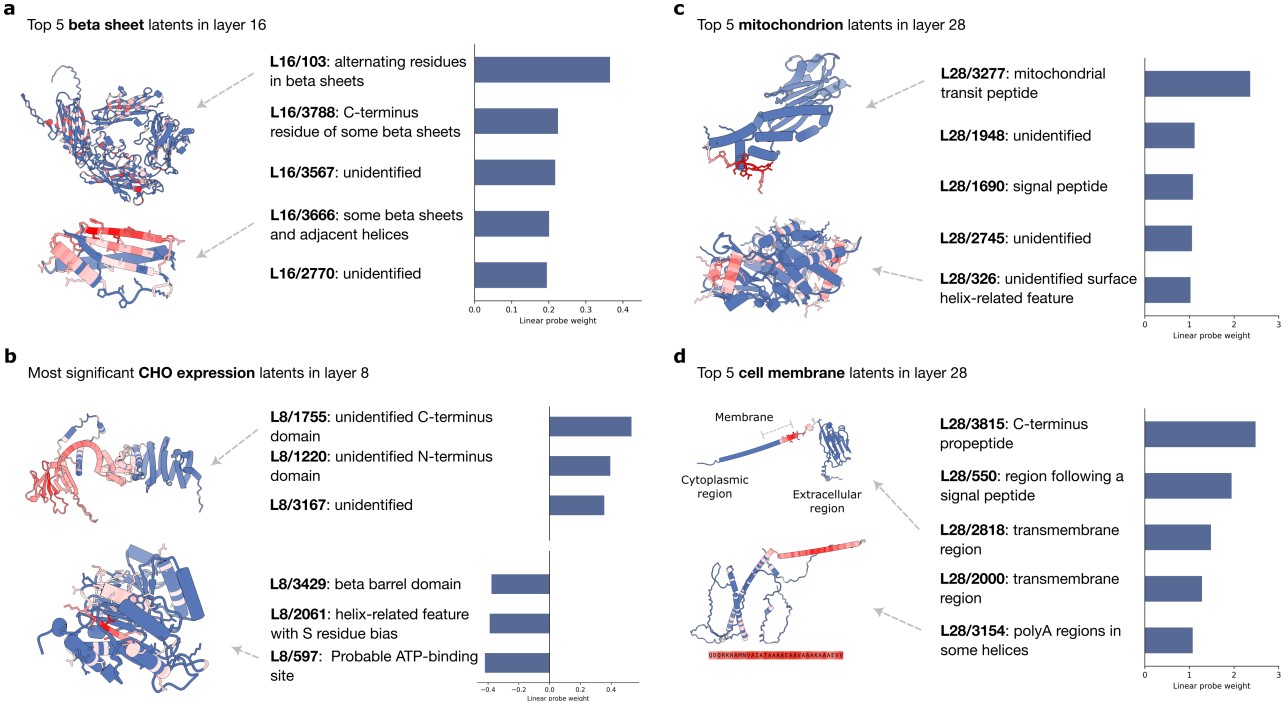

*Figure 9.* Linear probes identify predictive SAE latents for downstream tasks. **a**. The top 5 beta sheet latents from the secondary structure prediction task (SAE trained on ESM layer 16). **b**. The latents corresponding to the 3 most positive and 3 most negative coefficients from the linear probe on CHO cell line expression. The most positive latent activates on an unidentified motif; the most negative latent activates on an ATP binding site (SAE trained on ESM layer 8). **c-d**. Top 5 mitochondrion and cell membrane latents from the subcellular localization prediction task. The top nucleus latents identify transit peptides. The top cell membrane latents correlate with signal peptides and transmembrane regions (SAE trained on ESM layer 28).

## A.10. Distribution of linear probe coefficients

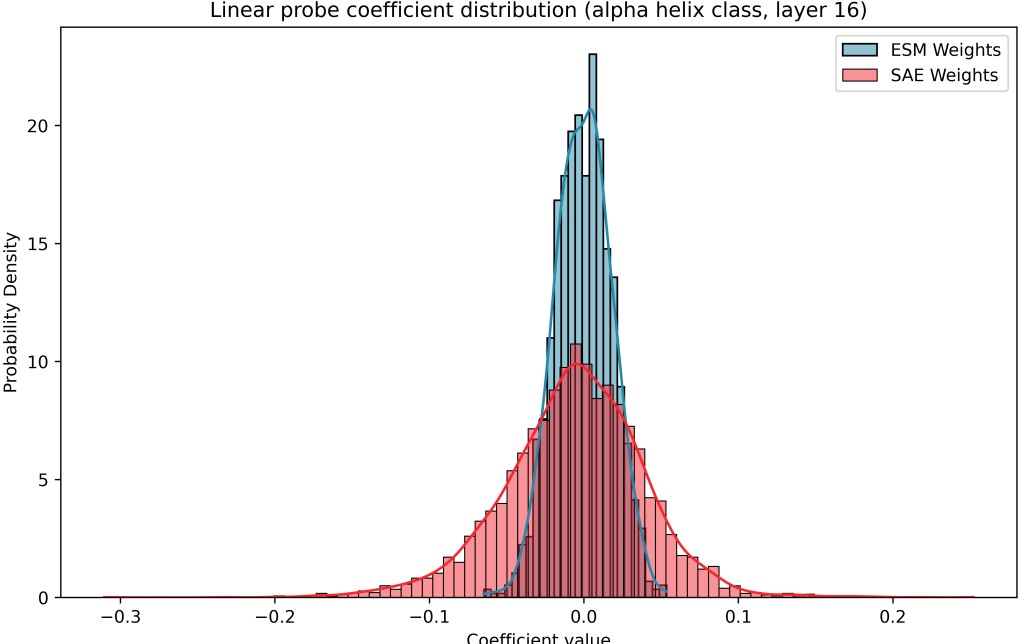

*Figure 10.* Example distribution of ESM vs. SAE linear probe coefficients. Shown here are coefficients for the alpha helix class from the secondary structure prediction task, using layer 16. The SAE coefficients have higher variance ($2.03\mathrm{e}{-3}$) than the ESM coefficients ($3.24\mathrm{e}{-4}$) because more latents are individually predictive of the label. The SAE coefficients also have a larger negative skew ($-2.05\mathrm{e}{-1}$) compared to ESM ($-6.51\mathrm{e}{-2}$). This can be explained by the SAE latents being more monosemantic: they sort more distinctly into helix vs. non-helix classes, and there are more non-helix examples than helix ones.

