# OpenReview forum: "From Mechanistic Interpretability to Mechanistic Biology: Training, Evaluating, and Interpreting Sparse Autoencoders on Protein Language Models"
_ICML.cc/2025/Conference — ICML 2025 spotlightposter_

### Official Review · Reviewer_fdKK · 2025-03-11

**Overall Recommendation:** 4

**Summary:**

This paper shows that SAEs can be used to better understand PLMs. They show that the features are interpretable via a series of case studies, including some clean histograms of activating examples. They use the SAEs to make high level observations about the PLMs, such as by categorizing the feature activation styles (point, short motif, etc), and studying the variation over the course of layers. Finally, they investigated probing for properties of proteins, using mean pooling, both on the residual stream and the feature activations. There was comparable performance suggesting that key information has not been lost by the SAE. Further, they find that the features used by a probe are often interpretable and find related concepts. Some of these relationships are not obvious but were discovered in prior research (?) suggesting SAEs have the potential to generate new hypotheses to validate.

**Claims And Evidence:**

Yes

**Essential References Not Discussed:**

No

**Experimental Designs Or Analyses:**

Yes

**Methods And Evaluation Criteria:**

Yes

**Other Comments Or Suggestions:**

## Major Comments
*The following are things that, if adequately addressed, would increase my score*

1. This was a delightful paper! I'm now fairly convinced that PLMs learn interpretable concepts and that SAEs have the potential to be a valuable tool for scientific discovery by extracting them. Interpretability for scientific discovery has long been a dream of the field, and this seems like a step in the right direction! This is a clear accept and I am open to a strong accept if the concerns below are addressed
    - Caveat: I work in interpretability, but not with PLMs, so my assessment of the novelty there is shaky. Ditto, I have not carefully checked the PLM related details of the work.
2. The main problem with the paper, in my opinion, is that it rests too much on qualitative case studies. These are great and add a lot of flavour and detail to the work, but are vulnerable to potential cherry picking. Assessing what I see as your key contributions for this:
3. Claim: SAE features are interpretable
    - This is OK. I think figure 3a is great, and I love the InterProt visualisations, but these are equally consistent with 5% of features being interpretable and 95% being
    - The main fix I would recommend is randomly choosing 100 features and for each one trying to interpret the top activating examples and just rating each for whether there's a consistent pattern. As done in eg the Gated Sparse Autoencoders paper
4. Claim: Features in different layers have different types of activation patterns
    - This seems well supported to me. Table 2 was very helpful and clear and seems reasonable. The area plot in 3c is great!
5. Probing
    - I struggled to follow the exact point you were trying to make here. The claims I took away from this were:
        - Probing still works on SAE activations, therefore they haven't broken anything or lost key info (but it need not be interpretable)
        - Given a labelled dataset for some concept, we can find related features by looking at large coefficients after training a probe
    - I found it a bit odd that you were using dense probes on the SAE activations, rather than sparse probes (which is what I consider to be standard - see eg Finding Neurons In A Haystack for discussion of different methods). But scoped into the two claims above this is OK, as the probe is just a method to find key latents
        - Note: I predict that the probing is unnecessarily complication and that you could find similarly good features by looking for those with the highest mean difference or difference in how often they activate between the positive and negative labels
    - It would be even better if you took random concepts (from some external list of suitable datasets, filtering for those that a dense linear probe works well for) and looked at the associated SAE latents, and said how often there's an interpretable connection (and how often it's trivial vs non-trivial)
6. I don't understand the "steering with a family feature changes it least at evolutionarily conserved points for that family". What exactly is the computation being done here? Shouldn't the pLM already be confident in the conserved points, so it's harder to shift? Have you compared to a control like steering with a feature of a different family?
    - An alternative experiment would be to look at the log prob of the relevant token and do gradient based attribution from each SAE feature to that log prob - ie, take the gradient of that log prob with respect to each feature, elementwise multiply by the activations, and add it up across the sequence dimension (basically a saliency map over SAE features). Family features should have high attribution in their families and not otherwise, according to this hypothesis. I recommend excluding the first token when doing this, as ablating SAE features is not a valid operation there (it's constant) and will often mess with your results
7. There may not be time in rebuttals, but I think your paper would be even stronger with causal evidence for the interpretable role of SAE features.
    - For example, showing that when you ablate a feature, the times when the correct next log prob decrease the most are interpretable
    - Or showing that you can steer the model in predictable ways by adding it in to unrelated contexts
    - One useful technique for exploring this would be gradient based attribution (approximating the effect of an ablation) of each SAE feature at each token to the next log prob at some interesting token - you could visualise this and see which tokens seemed most relevant and where, and if this matched your hypotheses
        - If you do this, be careful about hindsight bias! Eg do blinding, where you get shown 5 features, of which one had the highest attribution, and have to predict which is why

## Minor Comments
*The following are unlikely to change my score, but are comments and suggestions that I hope will improve the paper, and I leave it up to the authors whether to implement them. No need to reply to all of them in the rebuttal*

1. When, eg in Figure 3a, you argue that a feature has some explanation, you need to also check for false negatives - things in the beta-lactamase family that it doesn't fire on. Otherwise it could be much more specific, eg activating on every other token in beta-lactamase
    - This is a minor comment because basically every other SAE paper also fails on this point, and I expect it to in fact be more specific than that explanation
2. I am very familiar with SAEs but not PLMs. If you want people like me to be able to engage well with your work, having a short appendix primer on PLMs (or linking externally to one) defining key things would help a lot, eg what ESM-2 was trained on, how things are tokenized, what protein jargon like residues & secondary structures mean, etc.
3. If "quality of names" was a category, I'd give InterProt 5/5
4. I wonder if the several alpha helix features thing is feature splitting (as discussed in Towards Monosemanticity). You could test this by training an SAE with smaller dictionary size and seeing if an alpha helix feature forms
5. Mammalian cell expression seems very exciting - this doesn't change my assessment of the paper as it lacks any real exploration but I think it's reasonable to leave out of scope. I'd love to see the follow up paper though!

**Other Strengths And Weaknesses:**

See below

**Questions For Authors:**

See above

**Relation To Broader Scientific Literature:**

Covered well by the paper

**Theoretical Claims:**

N/A

---

> ### Author Rebuttal · Authors · 2025-04-01
>
> We thank the reviewer for their suggestions and enthusiasm for our work.
>
> We agree that one of the limitations of the paper is a heavy reliance on qualitative measures of feature interpretability. Towards making feature analysis more quantitative, we introduced the family specificity and activation pattern categorizations. In our experience, family specific-features are interpretable (can be interpreted as a specific sequence motif in a specific family), so the number of seemingly interpretable features is lower bounded by the number of family-specific features. But these measures admittedly don’t truly measure interpretability, and we would love to include results for human raters, and compare it to the ESM baseline. As done in the Gated SAE paper, we will conduct a blinded human rater experiment where our 5 raters who are familiar with protein biology will assess the interpretability of an SAE latent and ESM baseline as being interpretable (yes/maybe/no). We will include the results in a revision.
>
> Your interpretation of our goal in the probing section is correct; we are aiming to show that SAE representations haven’t lost key information (as demonstrated by task performance) and that the highest weighted latents correspond to features which makes biological sense with respect to the task. You are right that we could identify relevant features using a simpler method. Our motivation for probing, besides simply identifying relevant features for a task, was to demonstrate how SAE embeddings could be used as a drop-in replacement for dense ESM embeddings, without a significant hit to performance. Since in the pLM literature, linear probing is typically used to measure downstream task performance [1], we opted to do the same to make a head-on comparison.
>
> Regarding steering family specific latents, you make an excellent point. We will add a control where we steer a different (not family specific) latent to better demonstrate that family specific features are more important for conserved regions than other features. We will include this in the camera ready version. This is our attempt at getting more “causal” evidence to interpret SAE latents. As touched upon in our future directions section, getting interpretable steering results using SAE latents beyond simple amino acid specific features has remained challenging, and is a direction we will continue to explore in the future.
>
> In our calculation of the F1 score for family specificity, we also include sequences which do not activate the latent of interest. We believe this should help account for the false negative problem you raise.
>
> To make our work more accessible to the broader mechanistic interpretability, we will add a primer to introduce protein language models and some domain specific terms we often use in the paper. There have been some good reviews recently published which we can link as well [2]. And we’ve happy you like the name of our project!
>
>
> [1] https://www.biorxiv.org/content/10.1101/2024.02.05.578959v2
> [2] https://www.nature.com/articles/s41587-024-02123-4

---

> > ### Comment · Reviewer_fdKK · 2025-04-02
> >
> > Thanks! Those sound like great changes to the paper. I will maintain my score, but I think this is solid work that should obviously be accepted and plausibly spotlight

---

### Official Review · Reviewer_ZoZr · 2025-03-12

**Overall Recommendation:** 3

**Summary:**

The authors train SAEs on ESM-2 (a large protein language model), characterize the discovered features, use these organized features to better understand how ESM-2 learns protein representation.
They also develop a visualization tool, and find SAE features that correspond to known properties such as thermostability and subcellular localization.

**Claims And Evidence:**

Overall, the claims are well-supported by evidence.
I would prefer to shrink the claims from "pLMs" in general to ESM-2 unless the authors analyze more than one pLM (which they do not, as far as I can tell).
For example, "we determine that pLMs use a combination of generic features and family-specific features to represent a protein." (abstract, lines 024-026).

While I agree that SAEs are likely to apply to multiple pLMs, I think all SAE+pLM work has used ESM-2 (the authors note concurrent work, Simon and Zou 2024, also analyzes ESM-2). This is noted in the Limitations section already.

Minor: the authors claim that linear probes on SAE features are more reliably than linear probes on ESM features. This doesn't really appear true for Mammalian cell expression (Figure 4, lower right).

**Essential References Not Discussed:**

No essential references are missing.

**Experimental Designs Or Analyses:**

The experimental design is good.
In the related work, the authors argue that their work focuses on coevolution and scientific discovery.
I don't see any experiments focusing on coevolution; am I misunderstanding?

**Methods And Evaluation Criteria:**

The methods and evaluation are appropriate.
The authors are not proposing any new algorithmic innovation; they apply TopK SAEs to a pLM (ESM-2).
Their evaluations are qualitative in nature because they are interested in qualitative understanding of how useful SAEs are in the context of pLMs.

**Other Comments Or Suggestions:**

N/A

**Other Strengths And Weaknesses:**

This paper applies SAEs to pLMs.
It is one of the first works to do so.
This is undoubtedly a strength.

Unfortunately, the positioning (or lack thereof) makes it challenging for me (a non-biologist) to understand why the results are impactful.
I would improve my score if the authors can explain why the results are significant from a biology perspective.
Do SAEs offer a path towards answering questions that were previously unanswerable?
Are SAEs significantly cheaper, or more scalable, or easier to tune than other interpretability methods?
It's not clear to me why I should care about SAEs+pLMs---I'm sure there's a reason, but this current iteration does not convince me to care.

**Questions For Authors:**

N/A

**Relation To Broader Scientific Literature:**

The authors fairly position their work in the landscape of interpreting pLMs, using SAEs on pLMs, and dictionary learning methods on biological models.

However, it's unclear to me how SAEs improve on prior work.

**Theoretical Claims:**

There are no theoretical claims.

---

> ### Author Rebuttal · Authors · 2025-04-01
>
> We thank the reviewer for their detailed reading and thoughts.
>
> We acknowledge that the lack of other pLMs in our analysis beyond ESM-2 means that we should not make general conclusions about all pLMs. We intend to update the text to reflect this, replacing pLM with ESM-2 to narrow the scope of the results.
>
> As the reviewer points out, the performance of SAE probes on the mammalian expression task lags behind the ESM probes. In the text, we say that “linear probes on SAEs achieve performance similar to their ESM embedding baselines across all layers”. We intend to update the text to qualify this statement, that they achieve similar performance on most tasks.
>
> > In the related work, the authors argue that their work focuses on coevolution and scientific discovery. I don't see any experiments focusing on coevolution; am I misunderstanding?
>
> This is a good point, we could have been more specific. By coevolution, we refer to our analysis of features which seem to correspond to protein family-specific motifs.
>
> We would like to expand on why we view our results as significant from a biological perspective. Understanding biological sequences such as protein sequences remains a difficult problem. Traditionally, biologists have used tools such as multiple sequence alignments and statistical models to identify local patterns in protein sequences. Recently, protein language models trained on millions of sequences across evolution have been successful in providing useful protein representations. Embeddings from pLMs are often used as input to simple models (e.g. linear regression) on downstream tasks such as protein structure and variant effect prediction. Yet as these representations are uninterpretable, it is not easy to identify what features the pLM has learned which contribute to task performance. We advance on prior work by demonstrating SAEs can help reveal what ESM-2 learned features are important for downstream task performance. For example, by probing on SAE latents, we can identify that amino acid composition is an important predictor for thermostability. This may allow a biologist to gain insight into novel sequence determinants of a protein property which would have required manual feature crafting previously. As representations from ESM are widely used, we offer SAE representations as a drop-in replacement with (usually) similar performance as the ESM representations but with interpretable dimensions. We hope this clarifies the advantage of SAE-derived representations.

---

> > ### Comment · Reviewer_ZoZr · 2025-04-04
> >
> > Thank you for your further explanation. I think what I am still stuck on is language like this:
> >
> >     This may allow a biologist to gain insight into novel sequence determinants of a protein property which would have required manual feature crafting previously.
> >
> > It seems that after applying SAEs to PLMs, this work does has not demonstrated that there are new capabilities that were previously inaccessible without SAEs applied to PLMs.
> >
> > While I appreciate the technical difficulty of applying SAEs to PLMs and the novelty associated with being one of the first works to do so, I will leave my score as a 3 because I do not feel that your work demonstrates meaningful advances in capabilities. What does an SAE trained on a PLM actually unlock? I am, however, happy for this work to appear in ICML if other reviewers are excited about it.

---

### Official Review · Reviewer_qqyw · 2025-03-13

**Overall Recommendation:** 4

**Summary:**

The paper investigates the interpretability of protein language models by training sparse autoencoders on pLM latents (in particular from ESM2). The goal is to extract and analyze features that pLMs use to represent protein sequences, with the broader aim of linking these features to biological properties. The authors develop a visualization tool, InterProt, to examine the learned features and categorize them based on activation patterns and specificity to protein families. Theyfind that a large subset of latents are family-specific, but some are generic and pick things like intrinsically disordered regions, motifs and known heuristics (e.g. glutamine count for thermostability), down to single residues (prolines at helix boundaries). The authors train linear probes on SAE embeddings and compare their predictive performance to standard ESM embeddings across several downstream tasks. They demonstrate that SAEs can uncover biologically relevant features such as nuclear localization signals and thermostability determinants. The study also explores how SAE hyperparameters influence feature extraction, showing that increased sparsity leads to more family-specific features. Overall, the paper provides a framework for using SAEs to interpret pLMs and proposes that such models can facilitate biological discovery by identifying novel functional patterns in protein sequences.

**Claims And Evidence:**

The main claim is that pLMs do not merely memorize protein sequences but instead encode meaningful biological patterns. The authors argue that training SAEs on pLM activations allows them to reveal such patterns in a structured way which is easier to interpret. The claim that follows from the above is that these SAE-derived features are useful for biological discovery, as they can highlight functional determinants of protein properties that might otherwise be hidden in black-box model representations. The claims are reasonably well supported by experimental results: the authors train autoencoders on different layers of ESM-2 and develop the InterProt tool to inspect the learned features. They show that some latent dimensions correspond to biological concepts e.g. secondary structure, conserved sequence motifs, and biochemical patterns, and that many of these features activate within specific protein families. Furthermore it is shown that adjusting SAE hyperparameters (sparsity, expansion factor) influences the number of family specific features extracted.

Nevertheless, there are some remarks wrt the claims made:
1. As a result, the authors suggest pLMs encode biological knowledge beyond simple memorisation. In my opinion this is not entirely accurate. The interpretable latents are a compelling finding but e..g the presence of protein family specific latents could be interpreted as evidence of memorization (as opposed to true generalization) and so does not necessarily mean the model has learned fundamental biological principles rather than statistical correlations present in training data. Such limitation however, is alluded to by the authors in the discussion section.
2. Whether SAEs truly offer a systematic path to understanding pLMs or if they simply provide another layer of abstraction that still requires human intuition to decode remains to be determined not just in the biology domain but the wider ML space. Furthermore, their susceptibility to different hyperparameter settings is known (and something the authors also look at in the paper). The recent scrutiny by several works in the literature also calls them into question. I would have liked to see this addressed in a bit more detail with some further thought on other techniques from the mech interp literature that could be used for pLMs.

**Essential References Not Discussed:**

not applicable

**Experimental Designs Or Analyses:**

Experimental design and analysis are generally well structured and the experiments the authors choose to conduct make sense in this context. The only remark I have here relates to specific choices of hyper parameters in each setting. In this paper we are dealing with SAEs, which as mentioned earlier in this review, and as the authors themselves acknowledge, are sensitive to different hyperparameter choices. The authors do explore this direction a bit, but in my opinion could have done so in a bit more detail. The authors do not provide enough detail on hyperparameter selection strategy. Due to the biases that could be introduced by this step, this hinders reproducibility. The use of logistic regression for classification tasks and ridge regression for thermostability prediction is a reasonable choice, as these methods provide interpretable coefficients that can be analyzed to identify important features. The datasets used for evaluation are well-chosen and appropriate for the study. Secondary structure prediction, subcellular localization, thermostability, and mammalian cell expression are all biologically meaningful tasks. The classification of SAEs into different categories based on activation patterns and family specificity is useful for interpretability, but it is based on heuristics. The threshold of F1 > 0.7 for defining family-specific features is somewhat arbitrary.

**Methods And Evaluation Criteria:**

The methods and evaluation criteria are well matched to the problem. SAEs have been widely studied by the community in particular to interpret activations from LLMs, so their application to pLMs make sense in a similar fashion to how language modelling tasks were applied to protein sequences in the first place. Protein language models have shown strong predictive power for biological tasks but indeed their internal representations have not been as well studied, and this paper as well as a small number of recent works aim to fill this gap. Linear probing to evaluate feature importance is aligned to this goal from the perspective of identifying which features can be most predictive in several downstream tasks. One remark regarding mean pooling: in this case, mean pooling for protein level tasks can obscure important sequence specific details e.g. where functional determinants are localized to specific residues. While it simplifies model inputs, it may limit the ability of the probes to capture fine-grained sequence information. The authors do outline that there could be further work in this direction

**Other Comments Or Suggestions:**

not applicable

**Other Strengths And Weaknesses:**

One of the main strengths of this paper is its originality in applying SAEs to interpret pLMs. As mentioned earlier, this is a relatively new field and a great effort from the authors to test various properties of pLMs through this machinery. The introduction of InterProt as a tool for visualizing these learned features adds practical value. The paper is clearly written and well structured.

There is a concern with respect to novelty, as there have been other works from the literature doing several similar things, and by itself this work is not presenting a new method but applying known techniques to pretrained models. Furthermore, as mentioned earlier, the paper could benefit from more content and detail (in particular in the supplementary section) supporting their findings and the main thesis of the paper. There are some potential concerns with respect to statistical validation (e.g. F1 threshold) and the results would be stronger if there were quantitative comparisons between extracted features and known biological annotations. In the downstream tasks, evaluations demonstrate that SAE embedding are predictive but there is no ablation analysis to determine whether specific latents are directly responsible for performance improvements. Finally, there is some concern with respect to generalisability - the experiments focus on a single pLM (it is unclear what results other pLMs would yield) and dataset.

**Questions For Authors:**

not applicable

**Relation To Broader Scientific Literature:**

This paper builds upon research in protein language models and mechanistic interpretability. The field of mechanistic interpretability has seen increasing interest due to the advent of LLMs and techniques like such as sparse coding and dictionary learning have been applied to them to identify human interpretable features (see e.g. [1]). The idea of applying techniques from the interpretability literature to interpret pLMs is not in itself new but it is timely and a promising area of research gathering increasing interest by the community (see e.g. [2, 3, 4]).
Perhaps the most related work is InterPLM [3] which uses SAEs to identify human-interpretable features and correlating them with biological concepts such as binding sites, structural motifs and functional domains.

[1] https://arxiv.org/abs/2309.08600
[2] https://arxiv.org/abs/2411.06090
[3] https://arxiv.org/abs/2412.12101
[4] https://arxiv.org/abs/2502.09135

**Theoretical Claims:**

not applicable

---

> ### Author Rebuttal · Authors · 2025-04-01
>
> We thank the reviewer for the thorough review and many good suggestions.
>
> We agree with the reviewer’s points around the limitations of SAEs. The presence of a large number of family-specific features suggests that SAEs do indeed learn, or memorize, MSAs. Furthermore, the activation pattern of family-specific features support the hypothesis proposed by Zhang et al. [1] that pLMs store evolutionary statistics via common motifs. Although one cannot conclude from the analysis of SAE features that pLMs have learned any biophysics (as Zhang et al. suggest, they likely do not), the presence of non-family-specific features do present evidence that ESM2 has learned some patterns beyond memorizing common motifs within protein families. For example, features corresponding to generic helixes (L16/2611), hydrophobic core residues (L8/2064), and nuclear localization signals (L28/2375) suggest that pLMs do contain generic notions not specific to a protein family. Our mental model is that pLMs not only store co-evolutionarily information from MSAs but also some useful statistical interpolations between them.
>
> As is common in interpretable ML, understanding SAE latent representations still relies heavily on human intuition. In the case of pLM SAEs, we benefit from established biological knowledge—such as secondary structure and known motifs—which gives us useful reference points. It's encouraging that many SAE latents align with these expected features. However, our claims would be stronger with a quantitative comparison showing that SAE latents are more interpretable than an ESM baseline. To that end:
> - Though using a smaller model in the ESM family and with a different architecture, Simon et al. [2] showed that more SAE latents map to SwissProt annotations compared to ESM neurons. For a more comprehensive comparison, see our response to reviewer 5Na5.
> - In A.7., we show some analysis of linear probe weight distributions that weakly suggest SAE probe coefficients to be more interpretable.
> - As done in the Gated SAE paper, we will conduct a blinded human rater experiment where 5 raters who are familiar with protein biology will assess the interpretability of an SAE latent and ESM baseline as being interpretable (yes/maybe/no). We will include the results in a revision.
>
> Recent criticism of SAEspoint to the limitations of LLM-based auto-interpretation techniques [3], which were not used in our work. It has also been found that SAEs trained on the same data learn different features [15].
>
> Many other mechanistic interpretability techniques  have been applied to pLMs. For example, attention matrices have been shown to resemble pairwise contact maps ([4], [5], [6]), and contain information about motifs such as signal peptides [7], which are also represented as SAE features. More recent methods such as model diffing [8], trans-coders/cross-coders ([9], [10]), and other SAE architectures ([11], [12]) are promising future directions for this work.
>
> We agree that mean-pooling SAE embeddings when performing downstream probes has important limitations. For the presence of a short functional motif or residue-level signal, max-pooling will likely yield clearer results. We would also like to try more sophisticated pooling methods such as aggregating via optimal transport [14] in the future.
>
> For the definition of family-specific features, we set the F1 threshold to 0.7 because we observed there to be a significant dropoff in F1 score below that point. We will update the supplement with a figure demonstrating this.
>
> Finally, we agree with the suggestion to include more quantitative evaluations of SAE features. Doing so across different hyperparameter schemes would additionally enable a more principled hyperparameters selection strategy. Our initial hope was to achieve this via our feature categorization analysis but only observed small effects across different hyperparameters (A.4). We will evaluate our probes across SAEs trained with different hyperparameters. Specifically, a range of SAE latent hidden dimensions and a range of k values. We hope this will help demonstrate the robustness of our results.
>
> [1] https://tinyurl.com/m8a5ratb
> [2] https://tinyurl.com/3xsfefaj
> [3] https://arxiv.org/abs/2501.17727
> [4] https://arxiv.org/abs/2404.16014
> [5] https://arxiv.org/abs/2006.15222
> [6] https://tinyurl.com/yja6xyhf
> [7] https://tinyurl.com/mub4tuy5
> [8] https://tinyurl.com/4hwm4tp8
> [9] https://arxiv.org/abs/2406.11944
> [10] https://tinyurl.com/4hwm4tp8
> [11] https://arxiv.org/abs/2404.16014
> [12] https://arxiv.org/abs/2407.14435
> [13] https://tinyurl.com/yc2memtc
> [14] https://tinyurl.com/yc8j3fzt
> [15] https://arxiv.org/abs/2501.16615

---

> > ### Comment · Reviewer_qqyw · 2025-04-07
> >
> > Thank you for this detailed response. I am happy for this work to appear at ICML given the stated additions to the paper.

---

### Official Review · Reviewer_5Na5 · 2025-03-14

**Overall Recommendation:** 4

**Summary:**

This paper studies sparse autoencoders trained on the protein language model ESM-2. They find that the SAEs contain a variety of generic and family-specific features, as well as features that can be used to identify sequence determinants of properties such as thermostability and subcellular localization. They also provide an interactive visualization tool to help in the labeling or interpretation of SAE features. Finally, they explore the impact of SAE hyperparameters and ESM layers on the features learned by their SAEs.

**Claims And Evidence:**

The claims made in this paper are supported by clear evidence, with none being problematic. Please see more on the methods and evaluation criteria for parts that can be strengthened.

**Essential References Not Discussed:**

While this paper cites Simon & Zou (2024) as a concurrent work and notes the differences between the two works, it would be useful if more details were provided on the similarities and differences between the results of each paper, even if the models explored between the two are different. Did the authors of the other paper find similar breakdowns for each type of latent? Were there notable discrepancies between the conclusions drawn in the two papers? I believe further expanding on this discussion would strengthen this paper and highlight the potential generalizability and reproducibility of the claims made.

**Experimental Designs Or Analyses:**

The soundness and validity of all experimental analyses were checked, in particular for Sections 4.3, 4.4, and 5.2. Note that much of the analysis was highly qualitative in nature, and thus could not be rigorously checked without access to the proposed visualizer.

**Methods And Evaluation Criteria:**

The proposed datasets are intuitive for the task, however, the evaluation criteria is not very clear. In particular, given that one of the main claims of this paper is that SAEs can help to interpret pLMs, more could have been said about how interpretation was performed and the agreement between different labelers of the SAE features. Ideally, the proposed visualization tool, InterProt, would have been anonymously provided with the submission. Alternatively, some examples of most-activating sequences for a few different latents could be provided.

**Other Comments Or Suggestions:**

In the "future directions" section, the authors note that "training on task-relevant sequences may yield more function-specific latents." I would suggest they look at [1], which provides evidence for this behavior in task-specific SAEs.

[1] Makelov, Aleksandar, George Lange, and Neel Nanda. "Towards principled evaluations of sparse autoencoders for interpretability and control." arXiv preprint arXiv:2405.08366 (2024).

**Other Strengths And Weaknesses:**

Other strengths:
- The main strength of this paper is the ability to essentially create concept bottleneck models on top of pLMs by training probes on top of SAE features, thus being able to explain how pLMs are able to solve various downstream tasks and coming up with potentially hypotheses for the underlying mechanisms of those tasks.
- The proposed visualization tool also seems like a significant contribution of this work, but is currently unverified as it was not submitted with the paper for review.

Other weaknesses:
- It is not clear exactly what the use of SAEs unlocked in this paper. I think the authors could include more discussion of what new scientific understanding of protein modeling / pLMs was gained specifically as a result of using SAEs that would not have been found through rigorous evaluation of the base model or by probing it more traditionally.
- I wonder how SAEs compare against a prototype-based model, such as a ProtoPNet, that directly relies on canonical examples of features rather than learning relevant latent directions that must be labeled post-hoc to understand. While potentially outside the scope of this work, comparison against other methods would strengthen the work and provide stronger evidence for why SAEs are a reasonable and efficient method for understanding pLMs.

**Questions For Authors:**

N/A

**Relation To Broader Scientific Literature:**

The key contributions of this paper are mainly related to literature in mechanistic interpretability that explore how sparse autoencoders can be trained on LLMs and VLMs to understand the various features encoded by models. Given the growing popularity of SAEs, recent works have proposed applying them to scientific models, including similar concurrent work by Simon and Zou (2024) that trains on pLMs, SAE-Rad that trains on a medical VLM (Abdulaal et al, 2024), and more. By attempting to explain models in an unsupervised manner, SAEs have the potential to discover novel concepts that humans may not have been able to explain before. This paper validates the utility of SAEs/dictionary learning for understanding what features pLMs may be leveraging to perform downstream tasks, with novel findings on the differences between layers of pLMs and the organization and prevalence of various features. Thus, this paper's main contribution with respect to prior literature is evaluating if SAEs are interpretable or useful for this previously unexplored modality.

**Theoretical Claims:**

This work is mainly empirical, and thus no proofs were provided.

---

> ### Author Rebuttal · Authors · 2025-04-01
>
> We thank the reviewer for these thoughtful comments and suggestions.
>
> We created an anonymous link for our InterProt visualizer at http://icml.interprot.com and hope that it can provide context on our manual interpretation process and showcase some interpretable features. To support our claims around interpretability, we agree with the need for more human label data and quantitative analysis. As done in the Gated SAE paper (Rajamanoharan et al.), we will conduct a blinded human rater experiment where 5 raters who are familiar with protein biology will assess the interpretability of an SAE latent and ESM baseline as being interpretable (yes/maybe/no). We will include the results in a revision. We hope that this can provide more robust evidence on the value added by SAE compared to directly analyzing the base model.
>
> We agree with the lack of details on how our work differs from Simon et al. and plan to include more information in our revision. The key differences are as follows:
>
> - We trained TopK SAEs while Simon et al. used ReLU SAEs.
> - We used a larger, 650M param variant ESM-2, compared to the 8M variant used by Simon et al. The 650M param model is far more widely used.
> - We propose a framework for systematically categorizing latents by family-specificity and activation patterns; Simon et al. does not do latent categorization. We also evaluate the effect of different SAE hyperparameters on these feature categorizations.
> We use linear probes on four downstream tasks to extract interpretable features with the goal of enabling scientific discovery. Simon et al. focuses on demonstrating the feasibility of SAEs on pLMs and performs quantitative interpretability comparisons between ESM and SAE.
> - We share an open-source visualizer InterProt with data visualizations aimed for manual feature interpretation. For example, each SAE latent is displayed with a collection activating sequences across different activating ranges, whether they cluster within specific protein families, providing options to align them. InterProt also enables searching a sequence across all latents, a feature that has enabled discovery of interesting, interpretable latents starting from a protein of interest. Simon et al. also provides a visualizer though it focuses more on displaying activating distributions of each latent and whether it has been linked to any Swiss-Prot concepts.
> Simon et al. proposed a method to automate the interpretation of SAE features using an LLM. We did not explore this approach.
>
> We see models like ProtoPNet with built-in interpretability as an exciting and complementary direction to post-hoc interpretation (of usually larger and more performant pLMs) via SAEs. Concept bottleneck models have also been applied to proteins, and have shown competitive performance with pLMs [2]. While we agree that a comparison of SAEs to these methods is outside the scope of this work, we plan to add a discussion of these references to our Related Works section.
>
> Finally, we thank the reviewer for the reference on the previous work by Makelov et al. on the effects of training data distribution, a direction we plan to explore in follow-up work.
>
> [1] https://arxiv.org/abs/2404.16014
> [2] https://arxiv.org/abs/2411.06090

---

### Decision · Program_Chairs · 2025-05-01

**Decision:**

Accept (spotlight poster)

**Comment:**

This paper trains sparse autoencoders on a large protein language model (ESM-2), characterizes the discovered features, and uses these to better understand how ESM-2 learns protein representation. The authors also developed a visualization tool, and find features that correspond to known properties such as thermostability and subcellular localization. Overall, the claims are well-supported by evidence.
I agree with the reviewers that this is a solid work on a timely topic.